# Néel-type skyrmion in WTe$_2$/Fe$_3$GeTe$_2$ van der Waals heterostructure

Yingying Wu[1,9], Senfu Zhang [2,9], Junwei Zhang[2], Wei Wang[3], Yang Lin Zhu[4], Jin Hu [5], Gen Yin [1], Kin Wong [1], Chi Fang[6], Caihua Wan[6], Xiufeng Han [6], Qiming Shao[1], Takashi Taniguchi[7], Kenji Watanabe[7], Jiadong Zang[8], Zhiqiang Mao [4], Xixiang Zhang [2] & Kang L. Wang [1✉]

The promise of high-density and low-energy-consumption devices motivates the search for layered structures that stabilize chiral spin textures such as topologically protected skyrmions. At the same time, recently discovered long-range intrinsic magnetic orders in the two-dimensional van der Waals materials provide a new platform for the discovery of novel physics and effects. Here we demonstrate the Dzyaloshinskii–Moriya interaction and Néel-type skyrmions are induced at the WTe$_2$/Fe$_3$GeTe$_2$ interface. Transport measurements show the topological Hall effect in this heterostructure for temperatures below 100 K. Furthermore, Lorentz transmission electron microscopy is used to directly image Néel-type skyrmion lattice and the stripe-like magnetic domain structures as well. The interfacial coupling induced Dzyaloshinskii–Moriya interaction is estimated to have a large energy of 1.0 mJ m$^{-2}$. This work paves a path towards the skyrmionic devices based on van der Waals layered heterostructures.

[1] Department of Electrical and Computer Engineering, University of California—Los Angeles, Los Angeles, CA 90095, USA. [2] Physical Science and Engineering Division, King Abdullah University of Science and Technology, Thuwal 23955-6900, Saudi Arabia. [3] Key Laboratory of Flexible Electronics & Institute of Advanced Materials, Jiangsu National Synergetic Innovation Center for Advanced Materials, Nanjing Tech University, Nanjing 211816, China. [4] Department of Physics, Pennsylvania State University, University Park, PA 16802, USA. [5] Department of Physics, University of Arkansas, Fayetteville, AR 72701, USA. [6] Institute of Physics, Chinese Academy of Sciences, Beijing 100190, China. [7] National Institute for Materials Science, 1-1 Namiki, Tsukuba 305-0044, Japan. [8] Department of Physics and Astronomy, University of New Hampshire, Durham, NH 03824, USA. [9] These authors contributed equally: Yingying Wu, Senfu Zhang. ✉email: wang@ee.ucla.edu

Atomically thin, layered van der Waals (vdW) materials have been experimentally shown to host long-range magnetic orders recently[1,2], which could push the magnetic memory and information storage to the atomically thin limit and lead to ultra-compact next-generation spintronics[3,4]. Since the discovery of the ferromagnetism in $Cr_2Ge_2Te_6$[5,6], $CrI_3$[7–10], and $Fe_3GeTe_2$ (FGT)[11–14], such layered crystals have been at the frontier of material research. Besides the material itself, interfacial engineering in vdW heterostructures offers an effective methodology to spin-polarize or valley-polarize two-dimensional (2D) materials. Proximity effect from the interface has been widely researched for spin or valley polarization in 2D materials, like graphene on transition metal dichalcogenides[15–19] and $WSe_2$ on $CrI_3$[20–23]. Coupling 2D magnets to vdW materials not only lends the magnetic properties of these 2D magnets to the adjacent materials, but also modifies the magnetic properties of the 2D magnets themselves[24]. Among all the possible interfacial coupling for 2D magnets, spin orbit coupling proximity can play an important role when the atoms of 2D magnets are in contact of heavy elements, considering that the magnetic properties are intrinsically related to the spin orbit coupling.

Rashba spin–orbit coupling is known to lead to a strong Dzyaloshinskii–Moriya interaction (DMI) at the interface[24,25], where the broken inversion symmetry at the interface can change the magnetic states. DMI has been recognized as a key ingredient in the creation, stabilization, and manipulation of skyrmions and chiral domain walls. Whereas skyrmions from DMI become significant in the heavy metals/ferromagnet systems[26–28], there have been no direct observations of skyrmions in vdW heterostructures, even though the topological Hall effect has been reported in Cr-doped topological insulator (TI)/TI[29] and Mn-doped TI systems[30].

In this work, we have observed the topological Hall effect in $WTe_2$/FGT vdW heterostructures from transport measurements. More importantly, a large DMI energy of 1.0 mJ m$^{-2}$ has been determined in this system and the formation of Néel-type skyrmions has been captured with Lorentz transmission electron microscopy (L-TEM). The sizes of the directly observed skyrmions are ~150 nm at 94 K and ~80 nm at 198 K. This work helps promote 2D materials for ultra-compact spintronic devices.

## Results

**Thickness characterization of WTe₂ and FGT**. Before we show the proximity effect of the heterostructures, we first determine the properties of $WTe_2$ and FGT separately. The 1T' crystal structure of $WTe_2$ (Fig. 1a) has been confirmed in the previous work[31]. Atomic force microscopy was adopted to measure the thickness

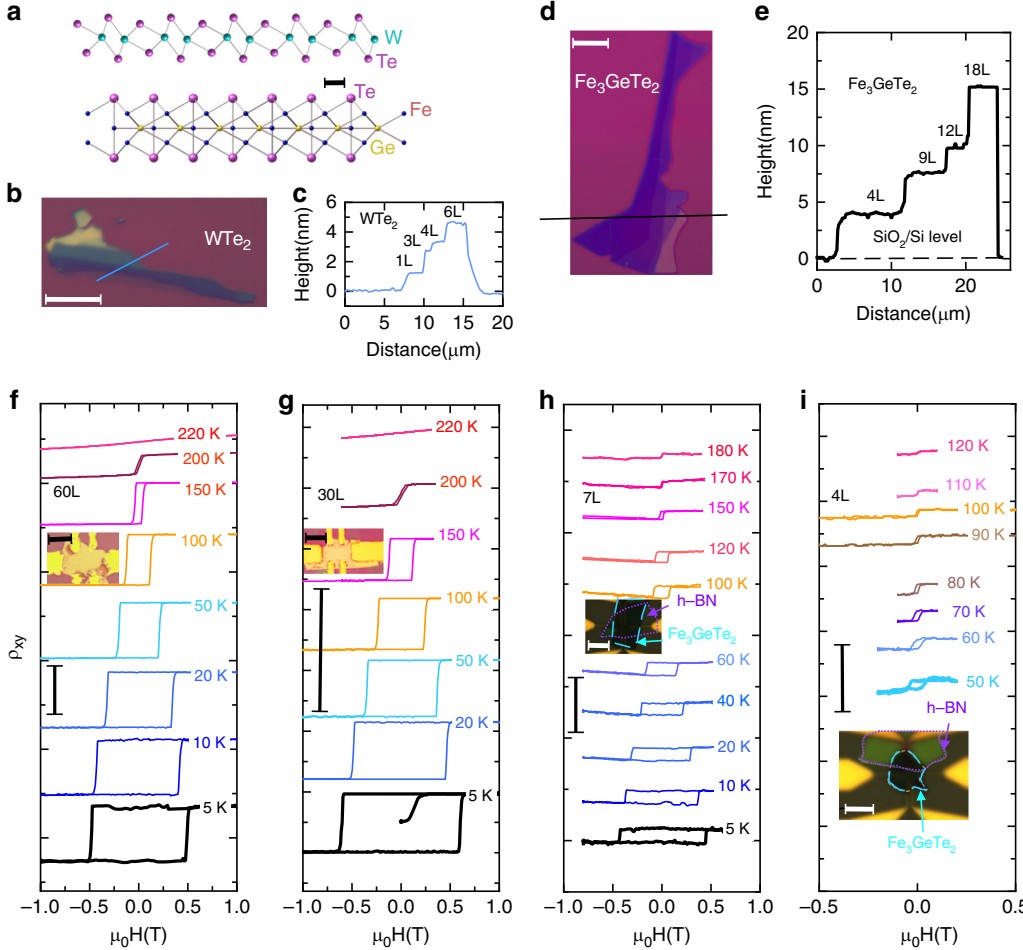

**Fig. 1 Thickness characterization and layer-dependent transport properties. a** Schematic graph for $WTe_2$ on $Fe_3GeTe_2$. Scale bar: 2 Å. **b** Microscopic image of exfoliated $WTe_2$ flakes. Scale bar: 10 μm. **c** Cross-sectional profile of the $WTe_2$ flakes along the blue line shown in (**b**). **d** Microscopic image of exfoliated $Fe_3GeTe_2$ thin films. Scale bar: 10 μm. **e** Cross-sectional profile of the $Fe_3GeTe_2$ flakes along the black line shown in (**d**). Temperature dependence of Hall resistivity for **f** 60L, **g** 30L, **h** 7L, and **i** 4L $Fe_3GeTe_2$ flakes showing that the Curie temperature decreases as the thickness of $Fe_3GeTe_2$ decreases. Insets show the devices for the measurements separately and the scale bar in the inset: 10 μm. Resistivity is shifted for clarity. The vertical scale bars are 10 Ω for (**f–h**) and 100 Ω for (**i**).

of this material as shown in Fig. 1b, c. It shows that the single-layer (1L) WTe$_2$ on the SiO$_2$/Si substrate has a height of about 1.1 nm. The transport data of 1L WTe$_2$ are given in Supplementary Note 1, Supplementary Fig. 1. Bulk FGT consists of weakly bonded Fe$_3$Ge layers that alternate with two Te layers with a space group P6$_3$ mmc as shown in Fig. 1a (bottom). For FGT, the 1L thickness is 0.8 nm[11]. This thickness is used to determine the number of layers for FGT as shown in Fig. 1d, e. As shown in Fig. 1f–i, the Curie temperature decreases from ~200 to ~100 K when the thickness of FGT goes down from 60L to 4L. The perpendicular magnetic anisotropy is well preserved for FGT with thickness down to seven layers. For the thin FGT samples with seven layers and four layers, hexagonal boron nitride (h-BN) thin flakes are used for protection from effects of the ambient conditions.

**Topological Hall effect in WTe$_2$/FGT heterostructure.** WTe$_2$ has one of the largest spin–orbit coupling among the transition metal dichalcogenides[32]. At the interface, the terminated atoms in WTe$_2$ are the same as those in FGT: two layers of heavy Te atoms are coupled through the 5p orbital, where the strong spin–orbit interaction from WTe$_2$ could play a significant role in reorganizing the spin polarizations in FGT. Such effect can be captured by transport measurements in thin films, and the transport data of h-BN/WTe$_2$/FGT heterostructures are shown in Fig. 2 with h-BN serves as the protection layer (Supplementary Note 2). For the h-BN/1L WTe$_2$/4L FGT sample (sample A) as shown in Fig. 2a, b, the longitudinal resistivity increases when temperature goes down from room temperature (Supplementary Note 3, Supplementary Fig. 2). The Curie temperature for this device is below 150 K, which is close to that of a 4L FGT on SiO$_2$/Si sample. Topological Hall is a spin-chirality-driven Hall effect, i.e., the spin chirality induces a finite contribution to the Hall response. In our case, dips and peaks near the magnetic phase transition edge, which signal the presence of the topological Hall effect, show up below 100 K as shown in Fig. 2c. For example, at 50 K there is a dip or peak at a field strength ~±975 Oe as indicated by the dashed lines (Supplementary Note 4, Supplementary Figs. 3 and 4). Different from this sample where the interfacial coupling has been reflected through the topological Hall effect, a 2L WTe$_2$/30L FGT heterostructure (sample B) shows the perpendicular magnetic anisotropy is well preserved in FGT when the temperature is below 180 K in the Fig. 2e, indicated by the square loop of the Hall resistivity. However, within an intermediate temperature range (180–200 K), the hysteresis loops differ in having transitions in steps. For example, at 190 K, the Hall resistivity suddenly jumps from the low saturation value to an intermediate level and then changes linearly. Finally, it saturates to the high saturation value at a relatively high positive field. The reversed trace does the opposite as expected. A possible explanation for this behavior is the formation of labyrinthine domain structures in FGT which results in multi domains[12], and this will be confirmed later. Compared to the sample A as shown in Fig. 2a–c, the sample B as shown in Fig. 2d, e with a much thicker FGT flakes measures the averaged transport signal which may lead to the smearing out of the topological Hall signal at the interface, which will be discussed later. In the following part, we utilize L-TEM to directly investigate the domain structures in 2L WTe$_2$/40L FGT and 2L WTe$_2$/30L FGT heterostructures, the latter of which is similar to sample B and we also confirm a large DMI at the interface.

**Experimental observation of Néel-type skyrmions with L-TEM.** Lorentz microscopy allows us to obtain direct magnetic domain structural images of thin magnetic films based on the fact that electrons experience a Lorentz force when traveling in a magnetic

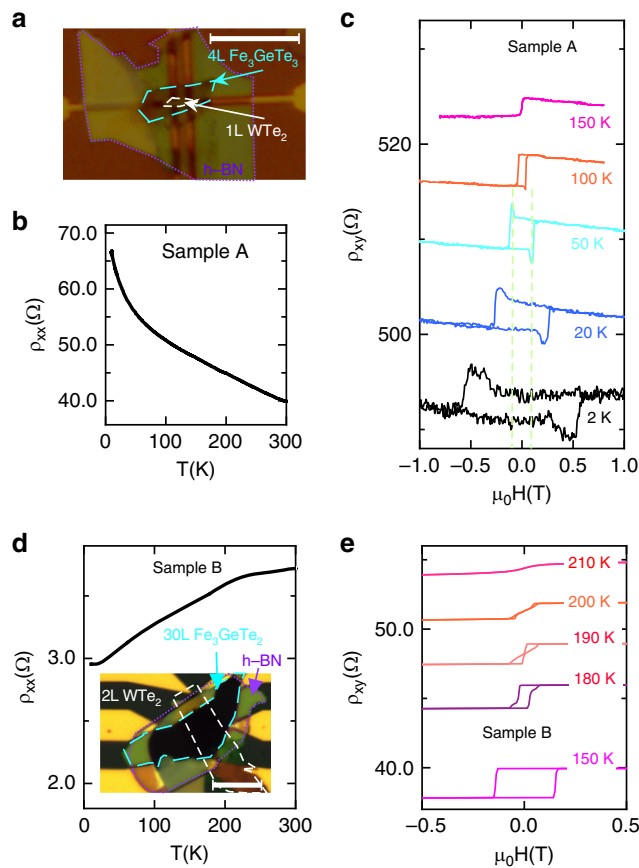

**Fig. 2 Transport properties of WTe$_2$/Fe$_3$GeTe$_2$ heterostructures.**
**a** Microscopic image of sample A (1L WTe$_2$/4L Fe$_3$GeTe$_2$). Scale bar: 10 μm. **b** Increasing longitudinal resistivity when temperature goes down for sample A. **c** Hall resistivity of the heterostructure shown in (**a**). Hall resistivity shows a peak and dip near the transition edge before the magnetization saturates, which is a sign of the topological Hall effect. An offset is used for clarity. **d** Longitudinal resistivity dependence on the temperature, showing the metallic behavior when temperature decreases. Inset shows the microscopic image of sample B (2L WTe$_2$/30L Fe$_3$GeTe$_2$). Scale bar: 10 μm. **e** Hall resistivity of the heterostructure shown in (**d**). An offset is used for clarity. Magnetic field is along out-of-plane direction.

field. Compared to other techniques[33], L-TEM, as one of the most direct methods to observe magnetic domain structures, domain walls and skyrmions, affords the advantage of a spatial resolution below 5 nm. The contrast formed in L-TEM is traditionally explained from the deflection of electrons due to the Lorentz force. For the Néel-type skyrmions, no contrast could be observed in the L-TEM image if the sample is not tilted because the intensity change caused by the Lorenz force at any point in the Néel-type skyrmion is compensated or canceled. After tilting the sample, dark-bright contrast could be formed in the L-TEM image[34] due to the contributions from both the outside and the core of the Néel-type skyrmion as schematically shown in Fig. 3a. The skyrmion lattice was observed in a 2L WTe$_2$/40L FGT heterostructure at 180 K for the rotating angle $\alpha = 30°$ (Fig. 3b) in de-focused and focused images. From under focus to over focus, the skyrmions transform from white on the top and dark in the bottom to the opposite contrasts. These are consistent with the Néel-type nature of the skyrmions (Supplementary Note 5, Supplementary Fig. 5). One hexagonal skyrmion lattice is indicated with the dashed lines. The field dependence of a single skyrmion in the WTe$_2$/30L FGT (similar to sample B for the transport

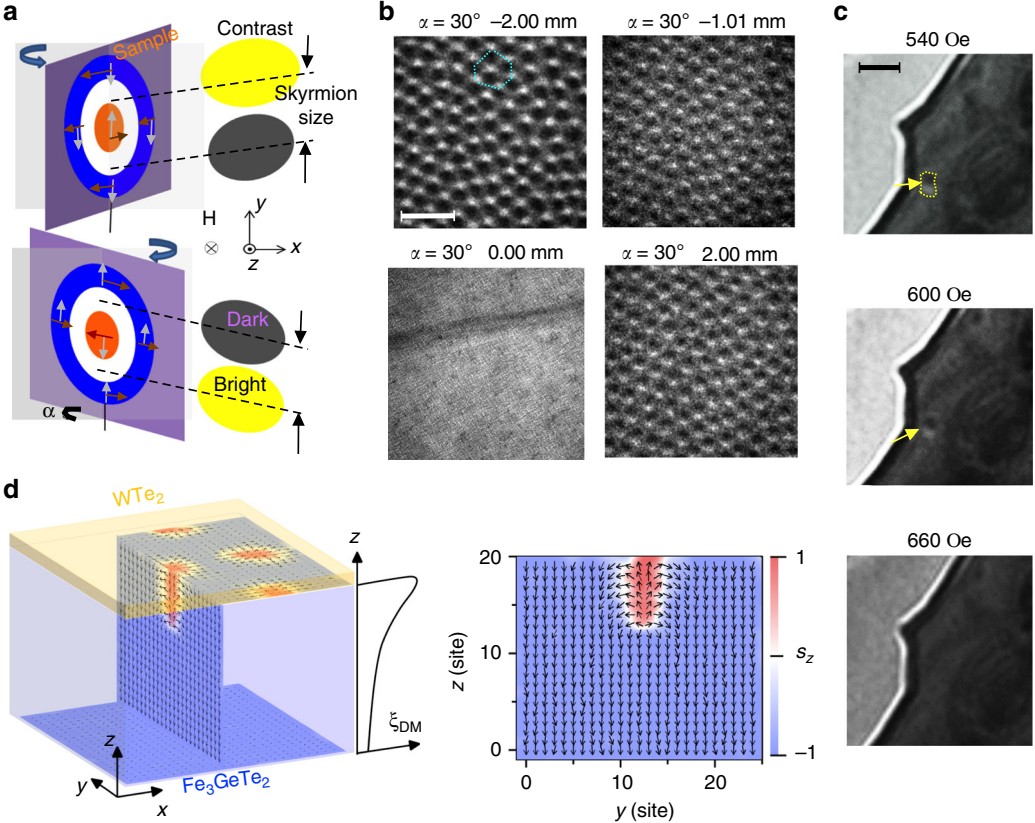

**Fig. 3 Néel-type skyrmions observed by Lorentz transmission electron microscopy. a** Schematic diagram of a Néel-type skyrmion on a tilt sample for Lorentz transmission electron microscopy imaging. The orange and blue circles are for positive and negative magnetizations along $z$ direction, respectively. Brown arrows indicate the in-plane magnetization component while gray arrows indicate the Lorentz force. **b** Lorentz transmission electron microscopy observation of skyrmion lattice from under focus to over focus on WTe$_2$/40L Fe$_3$GeTe$_2$ samples at 180 K with a field of 510 Oe. Scale bar: 500 nm. **c** Lorentz transmission electron microscopy observation of a Néel-type skyrmion at $T = 94$ K, $\alpha = 21.86°$ and $H = 540$, 600 Oe, where $\alpha$ is the angle between the sample plane and $xy$ plane. The yellow arrow points to a skyrmion. The skyrmion size is ~150 nm. Scale bar: 500 nm. **d** Simulation results considering a finite depth of interfacial Dzyaloshinskii–Moriya interaction in Fe$_3$GeTe$_2$.

measurements) is shown in Fig. 3c when a field changes from 540 to 660 Oe at 94 K. The Néel-type skyrmion is well developed at a field of 540 and 600 Oe along the $z$ direction, having dark on the top side and bright in the bottom. The size is estimated to be ~150 nm (more information about the skyrmion size of ~150 nm at 100 K and ~80 nm at 197 K can be found in Supplementary Note 6, Supplementary Fig. 6). However, we failed to resolve any domain structure in 1L WTe$_2$/4L FGT samples (similar to sample A) using L-TEM. This may be due to the small sheet magnetization in the thin film, which requires higher beam exposure to resolve; however, this already exceeds the tolerance of our samples, beyond which, unrecoverable damages occur. L-TEM measurements of WTe$_2$/FGT heterostructures with varied FGT thicknesses suggest the DMI only penetrates to a finite depth of FGT (Supplementary Note 7, Supplementary Fig. 7). To further understand how this interfacial DMI penetrates through the FGT layers, we have carried out simulations as shown in Fig. 3d. For FGT away from the interface, it enters the ferromagnetic phase as shown in the vertical profile in the $yz$ plane.

**Experimental evidence of WTe$_2$ induced DMI.** The strong perpendicular magnetic anisotropy favors out-of-plane magnetization, which makes the spin polarization in the FGT form the domain in the up or down directions. Figure 4b shows the magnetic domain images for the 30L FGT thin flake without WTe$_2$ (see Fig. 4a). The film exhibits labyrinth domains at 94 K and 0 Oe. When the in-plane field is tuned to the opposite

direction, the contrast of domain edge completely switches the sign as shown in the cases of a tilt angle of $\alpha = -20°$ and $\alpha = 21°$.

The formation and structure of domain walls are usually a result of the interplay between exchange interaction, magnetic anisotropy and dipolar interaction. Reducing the thickness of the FGT flakes decreases the dipolar interaction, thus, perpendicular magnetic anisotropy leads to a stabilized single domain[14]. Meanwhile, the exchange interaction term here includes DMI, which is a noncollinear exchange interaction that favors a chirally rotating magnetic structure of a specific rotational direction. By comparing the WTe$_2$/FGT and the nearby FGT regions, the magnetic domain difference shows the interfacial coupling contributes to a strong DMI.

For the WTe$_2$/FGT heterostructure, there is no contrast when $\alpha = 0°$ as shown in Fig. 4c, which is an evidence for the Néel-type domains. When $\alpha \neq 0°$, we have observed this well aligned and stripe-like domains with a domain width $w = 290$ nm at 0 T, which is sharply different from the domain structure of the FGT flake shown in Fig. 4b, having a much smaller domain width. In this case, the DMI is largely enhanced[28] for this heterostructure from the interfacial Te atoms coupling. A phenomenological model defines the dependence of the domain width $w$ on the domain wall energy $\delta_W$ by[35]

$$w = \beta \frac{4\pi \delta_W}{M_s^2}, \quad (1)$$

here the domain wall energy $\delta_W$ is related to exchange stiffness $A$, effective anisotropy constant $K_{\text{eff}}$, and DMI constant $|D|$ by

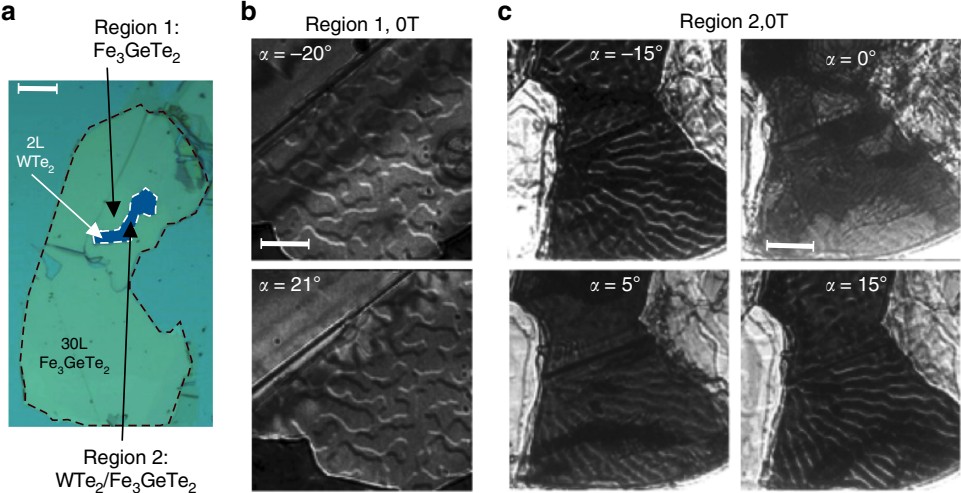

**Fig. 4 Magnetic domain difference of Fe₃GeTe₂ with and without WTe₂. a** Sample for Lorentz transmission electron microscopy measurements consisting of 2L WTe₂ and 30L Fe₃GeTe₂. Scale bar: 10 μm. **b** Typical labyrinth domain in 30L Fe₃GeTe₂ thin flakes. Scale bar: 2 μm. **c** From the aligned and stripe-like domain structures of the WTe₂/Fe₃GeTe₂, a Dzyaloshinskii–Moriya interaction energy is estimated to be ~1.0 mJ m$^{-2}$. Scale bar: 2 μm.

$\delta_W = 4\sqrt{A K_{\text{eff}}} - \pi|D|$[36]. $\beta$ is a phenomenological fitting parameter and taken to be 0.31 for FGT[35]. A domain width of 290 nm leads to a domain wall energy $\delta_w = 0.77$ mJ m$^{-2}$ including the DMI term (Supplementary Note 8, Supplementary Fig. 8). In the case of pristine FGT without considering DMI, the domain wall energy is simply expressed as $4\sqrt{A K_{\text{eff}}}$. Then by comparing these two domain wall energies with and without the DMI contribution, we obtain a DMI energy $|D| = 1.0$ mJ m$^{-2}$ in our system, which is comparable to the previous value in heavy metal/ferromagnet thin film systems[27] (Supplementary Note 9, Supplementary Fig. 9). One can compare $|D|$ to the critical value $|D_c|$, which is required to stabilize chiral Néel domain wall[27,37]

$$|D_c| = \frac{4}{\pi}\sqrt{\frac{A}{K_{\text{eff}}}K_d}, \qquad (2)$$

where when the magnetostatic or stray field energy constant $K_d = 2\pi M_s^2$ is large, the critical value for the Néel type domain is larger. Thus, $|D_c|$ is estimated to be ~0.1 mJ m$^{-2}$, so that $|D| > |D_c|$ and chiral Néel textures are expected. Compared to the transport measurements which take the averaged transport signal of the few nanometers of FGT coupled to the WTe₂ near the interface, and other FGT layers away from the interface, L-TEM helps confirm the DMI from the Néel-type skyrmion captured at the WTe₂/FGT interface since the uncoupled FGT layers in a single uniform domain contribute no contrast (Supplementary Note 10, Supplementary Figs. 10 and 11).

## Discussion

To bridge the gap of different thicknesses between the transport and the L-TEM studies, here we show two results suggest good consistency. On one hand, both skyrmion sizes obtained from the topological Hall effect and the L-TEM images give the same order of magnitude (Supplementary Note 11, Supplementary Fig. 12). On the other hand, the WTe₂ capping can only impact the domain structure for the FGT thickness less than 65L, suggesting a finite vertical penetration depth of the DMI. Assuming an exponential decay in the DMI profile, our simulation suggests that the skyrmions can only exist near the interface, where a large volume of the ferromagnetic phase shows up away from the interface (Supplementary Note 12, Supplementary Fig. 13). Due to frequent scatterings, when carriers are passing through the ferromagnetic phase,

they may quickly lose the memory of the transverse velocity due to the topological Hall effect, and therefore the anomalous Hall effect dominates. A quantitative analysis for the missing of the topological Hall effect in the thicker films is as following. Due to the large carrier density in thicker FGT films, it is expected to have a very small topological Hall signal. As an example, we estimate the topological Hall resistivity to be smaller than 0.01 Ω in WTe₂/30L FGT samples, which is more than two orders of magnitude smaller than the anomalous Hall resistivity. This may explain the missing topological Hall resistivity $\rho_{xy}^T$ humps in thicker films.

In summary, we reported the observed Néel-type skyrmions from L-TEM and the discovery of topological Hall effect in the WTe₂/FGT heterostructure. A large DMI energy of ~1.0 mJ m$^{-2}$ from the interfacial coupling may be a result of the broken inversion symmetry from Rashba spin–orbit coupling. We have shown Néel-type skyrmions of a small size (~150 nm at 94 K and ~80 nm at 198 K) at the interface of the vdW heterostructure. Further researches can include electrically gating to control skyrmions in 2D vdW heterostructures, and this may open a new area in the field of ultra-compact next-generation spintronics.

## Methods

**Sample assembly using a pick-up transfer technique**. In preparation, we coated the polydimethylsiloxane (PDMS) stamp on a glass slide with a polypropylene carbonate (PPC). During the assembly of heterostructure, we first exfoliated h-BN onto the 300-nm-thick SiO₂/Si substrate. Then the PDMS/PPC on the glass slide was used to pick up the h-BN from the SiO₂/Si substrate by heating up to around 40 °C. After that, we aligned the PDMS/PPC/h-BN and the exfoliated WTe₂ on the 300-nm-thick SiO₂/Si substrate, and used the similar heating temperature to pick up WTe₂. Then we used PDMS/PPC/h-BN/WTe₂ to pick up FGT. By heating the samples up to 110 °C, PPC was released from the PDMS and the PPC/h-BN/WTe₂/FGT was transferred onto the prepared bottom electrodes. Then acetone was used to remove the PPC thin film. Another way to assemble the samples is using the pick-up transfer method to form the PDMS/PPC/WTe₂/FGT layers first. After transferring the layers onto the bottom electrodes, the h-BN layers were used to cover WTe₂/FGT later, as shown in Supplementary Note 2. In both cases, the WTe₂/FGT interface is well preserved. The metal contacts used for the transport measurements of FGT thin flakes have two types, one is evaporation of Cr/Au electrodes after exfoliation of FGT onto the SiO₂/Si substrates and the other one is using the pick-up transfer technique to accurately align FGT with the prepared bottom Cr/Au electrodes. The length and width of contacts in a Hall bar geometry as shown in Fig. 1f–i for 60L FGT, 30L FGT, h-BN/7L FGT, h-BN/4L FGT are 15 μm × 9 μm, 15 μm × 9 μm, 3.5 μm × 0.5 μm, and 3.5 μm × 0.5 μm, respectively.

**L-TEM measurements**. The in situ L-TEM imaging was carried out by using a FEI Titan Cs Image TEM in Lorentz mode (the Fresnel imaging mode) at 300 kV.

## Data availability

The data that support the findings of this study are available in figshare with the identifier [https://doi.org/10.6084/m9.figshare.12520796].

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

## Acknowledgements

The sample synthesis and characterization efforts are supported by the US Department of Energy under grand DE-SC0019068. The transport measurements in this work are supported by the ARO program under contract W911NF-15-1-10561, National Science Foundation with Award nos. 1935362 and 1909416, the Spins and Heat in Nanoscale Electronic Systems (SHINES), an Energy Frontier Research Center funded by the US Department of Energy (DOE), Office of Science, Basic Energy Sciences (BES) under award #SC0012670. We are also grateful to the support from the National Science Foundation (DMR-1411085) and DOE, Office of Science, BES under Award No. DE-SC0020221. L-TEM measurements is based on research supported by the King Abdullah University of Science and Technology, Office of Sponsored Research and under the award No. OSR-2016-CRG5-2977.

## Author contributions

Y.W. and K.L.W. conceived the work and K.L.W. supervised the project. W.W. grew the FGT bulk crystals and Y.L.Z., J.H., and Z.M. grew the WTe₂ bulk crystals. G.Y. did the simulation. T.T and K.W. provided the h-BN crystal. Y.W. fabricated devices and performed the transport measurements. S.Z., J.Z., and X.Z. performed the Lorentz transmission electron microscopy measurements. K.W. and Y.W. carried out the atomic force microscopy measurements. C.F., C.W., and X.H. prepared the bottom electrodes. Y.W. analyzed the data with the help from Q.S and J.Z. Y.W. and K.L.W. wrote the paper with input from all authors.

## Competing interests

The authors declare no competing interests.
