## [Peer Review File · Nature Communications]

REVIEWERS' COMMENTS:

Reviewer #1 (Remarks to the Author):

The authors have addressed all of my recommendations satisfactorily. The revised manuscript and supplementary material contain more reproducible data and quantitative analysis. The text is better formulated for a general audience.

The only remaining weak point is that still there is no overlap in samples showing skyrmions in LTEM and samples showing topological Hall signal. The LTEM data can be taken as definite proof of skyrmions in the thicker samples. Combining this finding with the topological Hall signal in thinner samples makes the existence of skyrmions in thin samples very likely.

In my previous report I recommended "It would be good if the authors explicitly explains this difference between proven skyrmions in thick samples and likely skyrmions in thin samples." I did not find this explicit clarification in the new version of the manuscript.

Nevertheless, I recommend this article for publication.

Reviewer #2 (Remarks to the Author):

The authors have addressed most of my concerns, although some of the points are not treated well. I would like to recommend its publication of Nature Communications if and only if the following points are fixed.

1. The crystal structures of WTe₂ and FGT are still not correct. I would recommend using CrystalMaker or VESTA to write the precise crystal structure.
2. Please fix the tilted scalebar of Fig. S5.
3. Please put Fig. RII2 to the supplementary information, so that the readers can follow the evolution of the magnetic texture.

Point-to-Point Response on the Paper titled ‘Néel-type skyrmions in $\text{WTe}_2/\text{Fe}_3\text{GeTe}_2$ van der Waals heterostructure’

I. Response to referee #1

Comment: The authors have addressed all of my recommendations satisfactorily. The revised manuscript and supplementary material contain more reproducible data and quantitative analysis. The text is better formulated for a general audience.

The only remaining weak point is that still there is no overlap in samples showing skyrmions in LTEM and samples showing topological Hall signal. The LTEM data can be taken as definite proof of skyrmions in the thicker samples. Combining this finding with the topological Hall signal in thinner samples makes the existence of skyrmions in thin samples very likely.

In my previous report I recommended “It would be good if the authors explicitly explains this difference between proven skyrmions in thick samples and likely skyrmions in thin samples.” I did not find this explicit clarification in the new version of the manuscript.

Nevertheless, I recommend this article for publication.

Reply: We appreciate the referee’s point. We have to admit based on our existing probing methods, we cannot explicitly explain the difference between proven skyrmions in thick samples and likely skyrmions in thin samples. Currently, there are no WTe_2/FGT samples with the same thickness of FGT showing both the presence of skyrmions under L-TEM and topological Hall effect via the transport measurements. The small sheet magnetization in the thin film requires higher beam exposure to resolve in the L-TEM measurements; however, this already exceeds the tolerance of our sample, beyond which, unrecoverable damages occur. For the missing of topological Hall effect in the thicker films, it is mainly due to the large carrier density in thicker films, resulting in a smaller slope for ρ_{xy}^N , and thus a very small topological Hall signal. As an example, we estimated the topological Hall resistivity to be smaller than 0.01Ω in 30L FGT films coupled to WTe_2 . With advances in the techniques and more trials, we may be able to explain this explicitly in the near future.

II. Response to referee #2

Comments: The authors have addressed most of my concerns, although some of the points are not treated well. I would like to recommend its publication of Nature Communications if and only if the following points are fixed.

Reply: We thank the reviewer for all the important suggestions. In this new version of the manuscript, we have

- re-plotted the crystal structure of WTe_2 and FGT using CrystalMaker.
- fixed scalebar of Fig. S5, which is now Supplementary Figure 10 after revision.
- added Fig. RII2 to the Supplementary Information as Supplementary Figure 11.

Comments: 1, The crystal structures of WTe_2 and FGT are still not correct. I would recommend using CrystalMaker or VESTA to write the precise crystal structure.

Fig. RIII 1: Crystal structure from CrystalMaker. Scale bar: 2 Å

Reply: Thanks to the referee for this valuable point. We have replotted the crystal structures of WTe_2 and FGT using the CrystalMaker and combined them into the updated Fig. 1 in the main text.

Comments: 2. Please fix the tilted scalebar of Fig. S5.

Reply: Thanks referee for this important suggestion. We have fixed the tilted scalebar.

3. Please put Fig. RII2 to the supplementary information, so that the readers can follow the evolution of the magnetic texture..

Reply: We thank the referee for this point. Fig. RII2 now has been added into Supplementary Information as Supplementary Figure 11.